# Assessment of polygenic architecture and risk prediction based on common variants across fourteen cancers

Yan Dora Zhang ⓘ et al.#

Genome-wide association studies (GWAS) have led to the identification of hundreds of susceptibility loci across cancers, but the impact of further studies remains uncertain. Here we analyse summary-level data from GWAS of European ancestry across fourteen cancer sites to estimate the number of common susceptibility variants (polygenicity) and underlying effect-size distribution. All cancers show a high degree of polygenicity, involving at a minimum of thousands of loci. We project that sample sizes required to explain 80% of GWAS heritability vary from 60,000 cases for testicular to over 1,000,000 cases for lung cancer. The maximum relative risk achievable for subjects at the 99th risk percentile of underlying polygenic risk scores (PRS), compared to average risk, ranges from 12 for testicular to 2.5 for ovarian cancer. We show that PRS have potential for risk stratification for cancers of breast, colon and prostate, but less so for others because of modest heritability and lower incidence.

#A list of authors and their affiliations appears at the end of the paper.

Genome-wide association studies (GWASs) have led to the identification of hundreds of independent cancer susceptibility loci containing common, low-risk variants[1,2]. The number of discoveries varies widely across cancers, largely driven by available sample size, which reflects, in part, disease incidence in the general population. However, specific cancers, e.g., chronic lymphoid leukemia (CLL)[3] and testicular cancer[4], are notable for unexpectedly high numbers of genome-wide significant discoveries from GWASs of relatively small sample size. Previous studies have also reported that these two cancers have high heritability[5]. Across cancer types, polygenic risk scores (PRSs) show varying levels of risk stratification depending on the heritability explained by the identified variants and the disease incidence rates in the population[6–12]. Their potential clinical utility would depend not only on the level of risk stratification but also on other factors such as the availability of appropriate risk-reducing interventions for those identified as at high risk.

Estimation of heritability due to additive effects of all single-nucleotide polymorphisms (SNPs) included in GWAS arrays[13], referred to as GWAS heritability in this article, have shown that common variants have substantial potential to identify individuals at different levels of risk for many cancer types[14]. It remains, however, unclear how large the sample sizes of GWAS need to be to reap the full potential of PRS-based risk prediction. Herein we apply our recently published method[15] to estimate the degree of polygenicity and the effect-size distribution associated with common variants (minor allele frequency (MAF) > 0.05) across 14 different cancer types, based on summary-level association statistics from available GWASs[16–28] from populations of European ancestry (Supplementary Table 1). From these inferred parameters, we then provide projections of the expected number of common variants to be discovered and predictive performance of associated PRS as a function of increasing sample size for future GWASs. Finally, by incorporating age-specific incidence[29] from population-based cancer registries, we explore the magnitude of absolute risk stratification potentially achievable by PRS.

## Results

**Cancer polygenicity**. We found that cancers are highly polygenic, like other complex traits[15,30,31]. Estimates of the number of susceptibility variants with independent risk associations vary from ~1000 to 7500 between the 14 cancer sites (Table 1). For comparability, effect-size distributions are shown in groups of similarly sized GWASs with similar power for detecting associations (Fig. 1). For GWASs with <10,000 cancer cases (group 1), CLL and testicular cancer are each associated with 2000–2500 variants and characterized by a much larger proportion of variants with larger estimated effect sizes than for the other group 1 cancers, as reflected by wider effect-size distribution with heavier tails (Fig. 1, Table 1). GWAS heritability estimates indicate that, in aggregate, common variants explain a high degree of variation of risk for these two cancers. In contrast, in group 1, esophageal and oropharyngeal cancers are associated with a larger proportion of variants with substantially smaller effect sizes, compared with CLL and testicular cancers in group 1.

For GWASs with 10,000–25,000 cases (group 2), melanoma is noteworthy because it is associated with a wider effect size distribution than other group 2 cancers. The estimated number of susceptibility variants in this group ranges from 1000 to 2000. GWAS heritability estimates indicate that aggregated common variants make a relatively small contribution to ovarian and endometrial cancer susceptibility. Finally, for the 3 GWAS with >25,000 cases each (group 3), prostate cancer is remarkable for having more variants with large effect sizes, namely, the underlying effect-size distribution has a heavier tail, compared

with cancers of the breast and lung (Fig. 1). In this group, all three cancer types tend to have large numbers of associated variants (>4500) compared with cancer sites in other groups, but this pattern could partially be due to the very large sample sizes of group 3 GWAS[15].

For a large majority of the 14 cancer sites, a two-component normal-mixture model for non-null effects provides a substantially better fit to observed summary statistics than a single normal distribution; this indicates the presence of a fraction of variants with distinctly larger effect sizes than the remaining (Supplementary Figs. 1 and 2). In contrast, a single normal distribution appears to be adequate for esophageal and oropharyngeal cancer, indicating the presence of a large number of variants with a continuum of small effects, similar to our previous findings for traits related to mental health and abilities[15]. Across all 14 cancers, the predicted number of discoveries and their associated genetic variance explained for current GWAS sample sizes match well to those observed empirically (Supplementary Table 2), indicating good fit of our model to the observed data.

**Future GWAS projections**. GWAS heritability estimates indicate that the potential of PRS for risk discrimination in the population varies widely among cancer types (Table 1). The area under the curve (AUC) statistics associated with the best achievable PRS varies from 64% (endometrial and ovarian cancer) to 88% (testicular cancer) and in the range of 70–80% for most cancers. The percentage of GWAS heritability explained by known variants varies widely, depending on study sample size and the underlying trait genetic architecture (Fig. 2). Known variants explain more than a quarter of heritability for cancer sites based on very large sample sizes (e.g., breast and prostate cancer) or for cancer sites that have susceptibility variants with relatively large effect sizes (e.g., CLL, melanoma, and testicular cancer). Oropharyngeal cancer, in contrast, has both a small sample size and small effect sizes; its percentage heritability currently explained is almost zero.

The sample size needed to identify common variants that could explain approximately 80% of the total GWAS heritability for the cancers evaluated is generally very large, requiring 200,000–1,000,000 cancer cases, with a comparable number of controls (Fig. 2). However, for three sites, namely, testicular cancer, CLL, and melanoma, the required sample size is smaller, 60,000, 80,000, and 110,000 cases, respectively, due to the large effect sizes of their associated variants. By quadrupling the sample sizes of currently published GWASs, the percentage of GWAS heritability explained would rise to >40% across all cancers, except for oropharyngeal cancer. Such sample size increases would also lead to appreciable improvements in PRS discriminatory power across all these sites (Figs. 3 and 4). For cancers that were found to be the most polygenic and that had small effect sizes (e.g., cancers of breast, lung, and oropharynx), improvement would occur at a slower rates as sample sizes increase, and these sites would require the largest sample sizes to generate PRSs with discriminatory power close to theoretical limits. Of note, for a number of cancers, the achievable relative risks for subjects at the 99th percentile of PRS distribution compared with those at average risk, are comparable to those for monogenic disorders[32] (e.g., relative-risk >3–4-fold) (Fig. 4). Across all 14 cancer types, inclusion of SNPs using more liberal but optimized $p$ value thresholds (see "Methods") would improve performance of PRS-based risk prediction versus using the stringent genome-wide significance level, but the anticipated gains would be generally modest (Supplementary Figs. 3 and 4).

Projections of residual lifetime cancer risks for the US non-Hispanic white population show that the discriminatory power of PRS built from current or foreseeable studies will depend heavily

**Table 1 Estimated number of independent common susceptibility variants and heritability across 14 cancer sites.**

| Number of cases in the analysis | Cancer site[a] | Total number of susceptibility SNPs (SE) | Total heritability, in log-OR scale[b] (SE) | Average heritability explained per susceptibility SNP[c] (SE), in $10^{-4}$ | Number of SNPs associated with larger variance component (SE) | % of heritability explained by SNPs with larger variance component | AUC associated with the best PRS[d] (SE) |
|---|---|---|---|---|---|---|---|
| <10,000 | CLL | 2025 (1501) | 1.62 (0.37) | 7.2 (4.4) | 52 (15) | 41 | 0.82 (0.03) |
| <10,000 | Esophageal | 3641 (2515) | 1.24 (0.36) | 3.4 (1.9) | NA[e] | NA | 0.78 (0.03) |
| <10,000 | Testicular | 2598 (2088) | 2.81 (0.40) | 9.2 (6.6) | 196 (75) | 54 | 0.88 (0.02) |
| <10,000 | Oropharyngeal | 3623 (2060) | 0.68 (0.27) | 1.9 (0.5) | NA | NA | 0.72 (0.04) |
| <10,000 | Pancreas | 1757 (1490) | 0.60 (0.16) | 3.2 (2.2) | 47 (27) | 31 | 0.71 (0.03) |
| 10,000–25,000 | Renal | 2220 (1555) | 0.57 (0.12) | 2.4 (1.4) | 46 (36) | 24 | 0.70 (0.02) |
| 10,000–25,000 | Glioma | 2364 (1593) | 0.87 (0.11) | 2.2 (1.2) | 61 (25) | 55 | 0.75 (0.01) |
| 10,000–25,000 | Melanoma | 1098 (533) | 0.65 (0.09) | 4.4 (1.6) | 106 (58) | 52 | 0.72 (0.01) |
| 10,000–25,000 | Colorectal | 1484 (696) | 0.43 (0.10) | 2.9 (0.8) | 14 (11) | 7 | 0.68 (0.02) |
| 10,000–25,000 | Endometrial | 1052 (772) | 0.27 (0.07) | 2.5 (1.3) | 46 (34) | 26 | 0.64 (0.02) |
| 10,000–25,000 | Ovarian | 1015 (715) | 0.24 (0.06) | 2.2 (1.1) | 49 (31) | 36 | 0.64 (0.02) |
| >25,000 | Lung | 6096 (2750) | 0.39 (0.06) | 0.6 (0.2) | 15 (7) | 15 | 0.67 (0.01) |
| >25,000 | Prostate | 4530 (1052) | 0.77 (0.04) | 1.1 (0.2) | 276 (99) | 51 | 0.73 (0.01) |
| >25,000 | Breast | 7599 (1615) | 0.60 (0.03) | 0.6 (0.1) | 587 (133) | 56 | 0.71 (0.00) |

*SNP* single-nucleotide polymorphism, *SE* standard errors, *CLL* chronic lymphocytic leukemia.
[a]All results are reported using the best fitted (two- or three-component) normal mixture model for effect-size distributions, with respect to a reference panel of 1.07 million common SNPs included in the Hapmap3 panel after removal of MHC region.
[b]Total heritability is characterized by population variance of the underlying true PRS as $h^2 = Var\left(\sum_{m=1}^{M} \beta_m G_m\right) = M\pi_c E(\beta^2)$, where $E(\beta^2)$ denotes per-SNP effect-size of the non-null SNPs in the log-odds-ratio scale.
[c]Average heritability explained per susceptibility SNP excludes SNPs with extremely large effects (see "Methods").
[d]Area under the curve (AUC) associated with best PRS is calculated using the formula AUC=$\Phi(\sqrt{h^2/2})$ where $\Phi(\cdot)$ is the cumulative density function of standard normal distribution.
[e]NA indicates that a two-component model is favorable compared to three-component model.

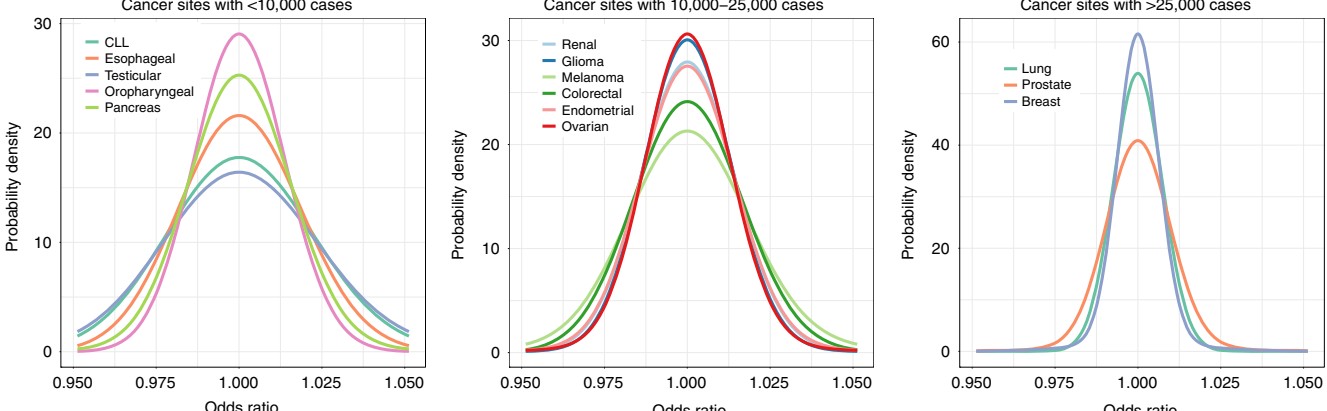

**Fig. 1 Estimated effect-size distributions for susceptibility SNPs across 14 cancer sites.** Effect-size distribution of susceptibility SNPs is modeled using a two-component normal mixture model for all sites, except esophageal and oropharyngeal cancers. For these sites, effect sizes are modeled using a single normal distribution that provided similar fit as the two-component normal mixture model (see Supplementary Figs. 1 and 2). SNPs with extremely large effects are excluded for effect-size distribution estimation (see "Methods"). Plots are stratified by sample size of the GWAS for comparability. Distributions with fatter tails imply the underlying traits have relatively greater number of susceptibility SNPs with larger effects. Note here that the effect-size distribution is plotted on the log scale of odds ratio (x-axis). CLL chronic lymphocytic leukemia.

on the underlying cancer incidence in the population (Fig. 5, Supplementary Figs. 5–7). The potential clinical utility of PRS depends on the degree of risk stratification and specific prevention or early detection strategies for a given cancer, should they exist. For common cancers, such as breast, colorectal, and prostate, a PRS with even modest discriminatory power (maximum AUC of approximately 70%, Fig. 3) can provide substantial stratification of absolute risk in the population. In contrast, for CLL and testicular cancer, even though its PRS could achieve a higher AUC (e.g. in the range 80–90%, Fig. 3), the degree of absolute risk stratification will be modest because of the infrequency of these cancers. Thus a PRS by itself has the least

impact on risk stratification for cancer sites that are infrequent or/and that have low heritability. However, it is possible that PRS could have clinical utility for some of these cancers in the presence or in combination with other risk factors and biomarkers. For example, a PRS for lung cancer may provide larger stratification for absolute risk among smokers than never smokers because of the higher baseline risk in smokers.

## Discussion

Our study is subject to several limitations. We may have underestimated the number of underlying common susceptibility loci,

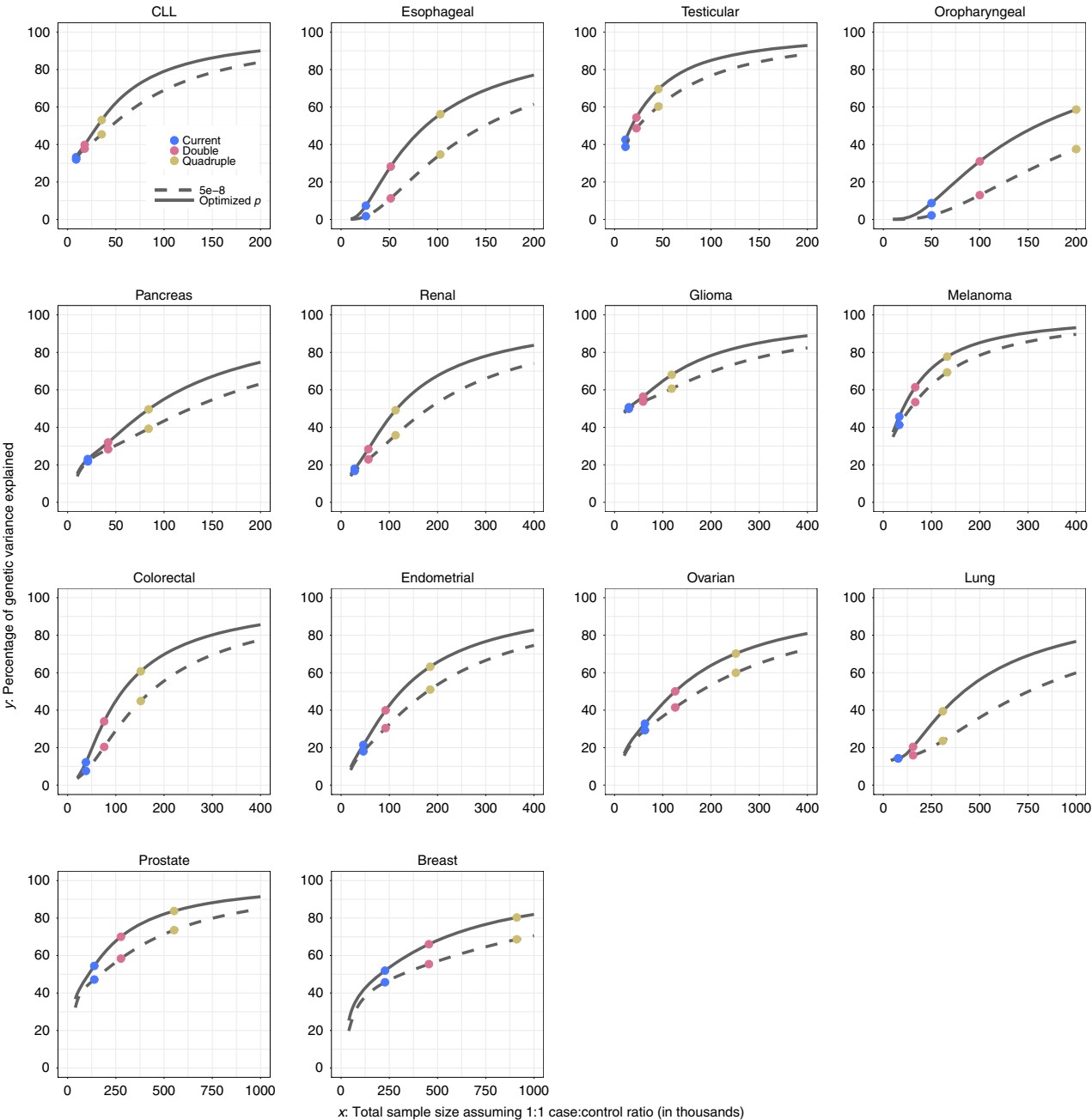

**Fig. 2 Projections of percentage of GWAS heritability explained by SNPs as sample size for GWAS increases.** Results are shown for projections including SNPs at the optimized $p$ value threshold (solid curve) and at genome-wide significance ($p < 5 \times 10^{-8}$) level (dashed curve). Colored dots correspond to sample size for the largest published GWAS and those for doubled and quadruped sizes. For oropharyngeal cancer, the projections at the "current sample size" are based on a sample size of 25K cases and 25K controls. For breast and esophageal cancer, the projections at the "current sample size" are based on the current largest GWAS sample sizes: 123K cases and 106K controls and 10K cases and 17K controls, respectively. For all other cancer sites, the projections at the "current sample size" are based on the GWAS sample sizes in Supplementary Table 1. CLL chronic lymphocytic leukemia.

especially for those cancers for which current GWAS have small sample sizes[15]. Thus the interpretation of comparisons of the underlying genetic architecture across cancer types with very different sample sizes requires caution. Nevertheless, the major patterns are unlikely to be due to differences in sample size. For example, we estimated oropharyngeal and esophageal cancers to be two of the most polygenic sites, though the GWAS sample sizes for these two sites were relatively small. Further, Q–Q plots of observed and expected $p$ values indicate that the inferred models for effect-size distributions explain observed GWAS summary statistics well, regardless of GWAS sample size.

Another important limitation is that we only included data from subjects of European ancestry, since GWAS data for other ancestries are currently too small to permit reliable projections for most cancer sites. In addition, several cancers (e.g., lung, ovary, glioma, and breast) consist of etiologically heterogeneous subtypes that were not considered in our analyses due to lack of adequate sample sizes for appropriate subtypes for most of these cancer sites. Further studies of ancestry- and subtype-specific genetic architectures are needed to address these limitations.

In our projections, we assume standard agnostic association analysis of SNPs without incorporating any external information

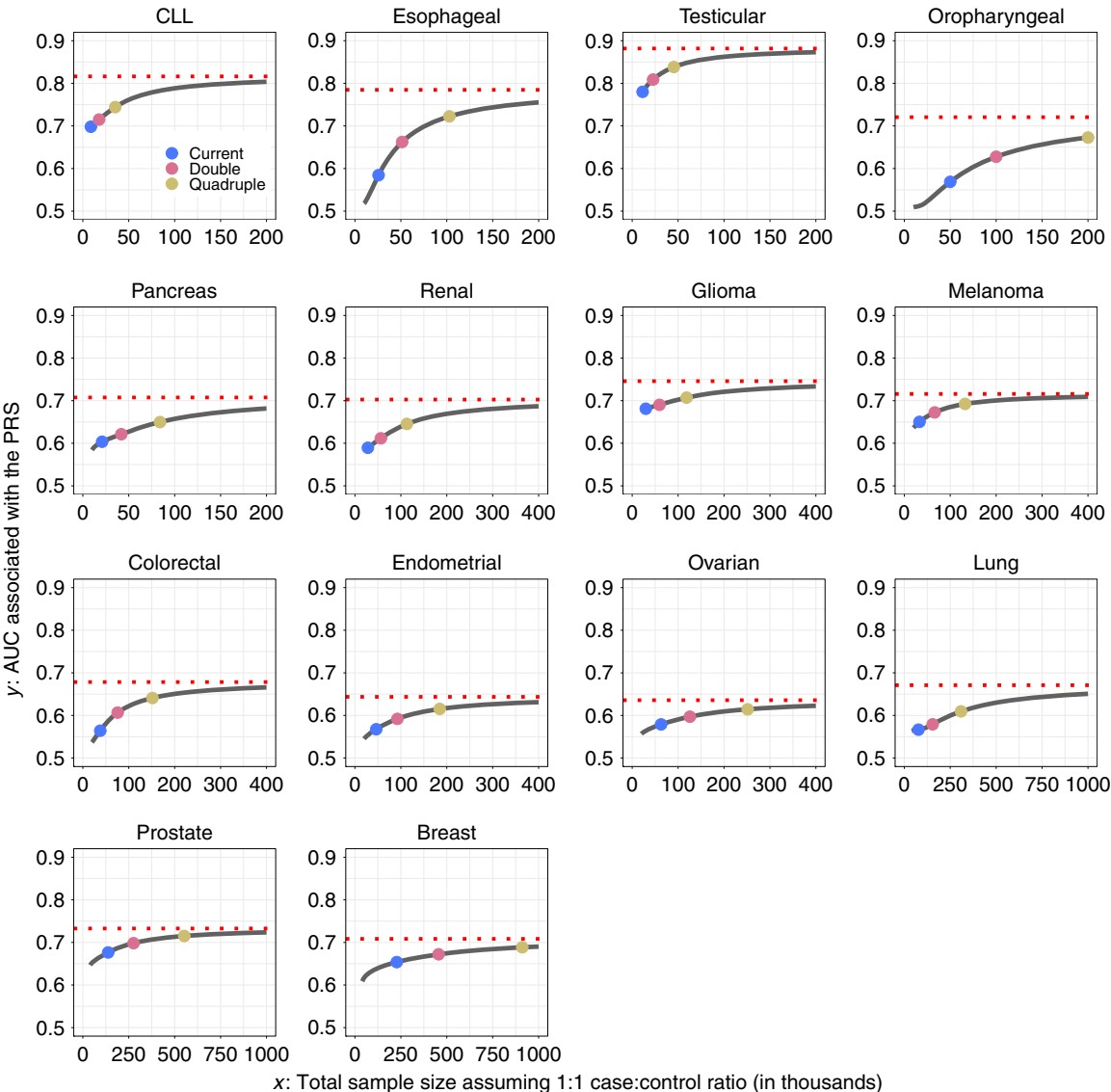

**Fig. 3 Projections of area under the curve (AUC) characterizing predictive performance of PRS as sample size for GWAS increases.** Results are shown for PRS including SNPs at the optimized *p* value threshold. The dotted horizontal red line indicates the maximum AUC achievable according to the estimate of GWAS heritability. Colored dots correspond to sample size for largest published GWAS and those for doubled and quadruped sizes. For oropharyngeal cancer, the projections at the "current sample size" are based on a sample size of 25K cases and 25K controls. For breast and esophageal cancer, the projections at the "current sample size" are based on the current largest GWAS sample sizes: 123K cases and 106K controls and 10K cases and 17K controls, respectively. For all other cancer sites, the projections at the "current sample size" are based on the GWAS sample sizes in Supplementary Table 1. CLL chronic lymphocytic leukemia.

on population genetics or functional characteristics of SNPs. It is, however, possible to incorporate various types of external information to improve power for discovery of associations[33–36] and genetic risk prediction[37]. We have evaluated the merit of future GWAS only in terms of their ability to explain heritability and improve risk prediction. However, current and future discoveries have other major implications, including provident insights to biological pathways and mechanisms, potential gene–environment interactions, and understanding causal relationships through Mendelian Randomization analyses[38]. A number of these cancers are known to have rare high-penetrant risk variants, but for this study we have focused on estimating effect-size distribution associated with common variants. Furthermore, heritability analysis indicate that uncommon and rare variants could explain a substantial fraction of the variation of complex traits[39], and thus it is likely that there are many

unknown uncommon and rare variants associated with these cancers as well. In the future, characterization of heritability and effect-size distribution associated with the full spectrum of allele frequencies will require individual-level sequencing data on a substantially larger number of cases and controls.

The observed differences in the underlying genetic architecture of susceptibility across cancers could be due to various factors, including the effect of negative selection[30,40], tissue-specific genetic regulation of gene expression[41], cell of origin[42], the number of biological steps needed to transition from normal to malignant tissue[43], mediation of genetic effects by underlying environmental exposures[44], and the presence of heterogeneous cancer-specific subtypes[21,25,27,28]. A number of cancer types, including those of lung, oropharynx, and esophagus, which were associated with large numbers of SNPs with small average effect sizes, have known strong environmental risk factors and distinct

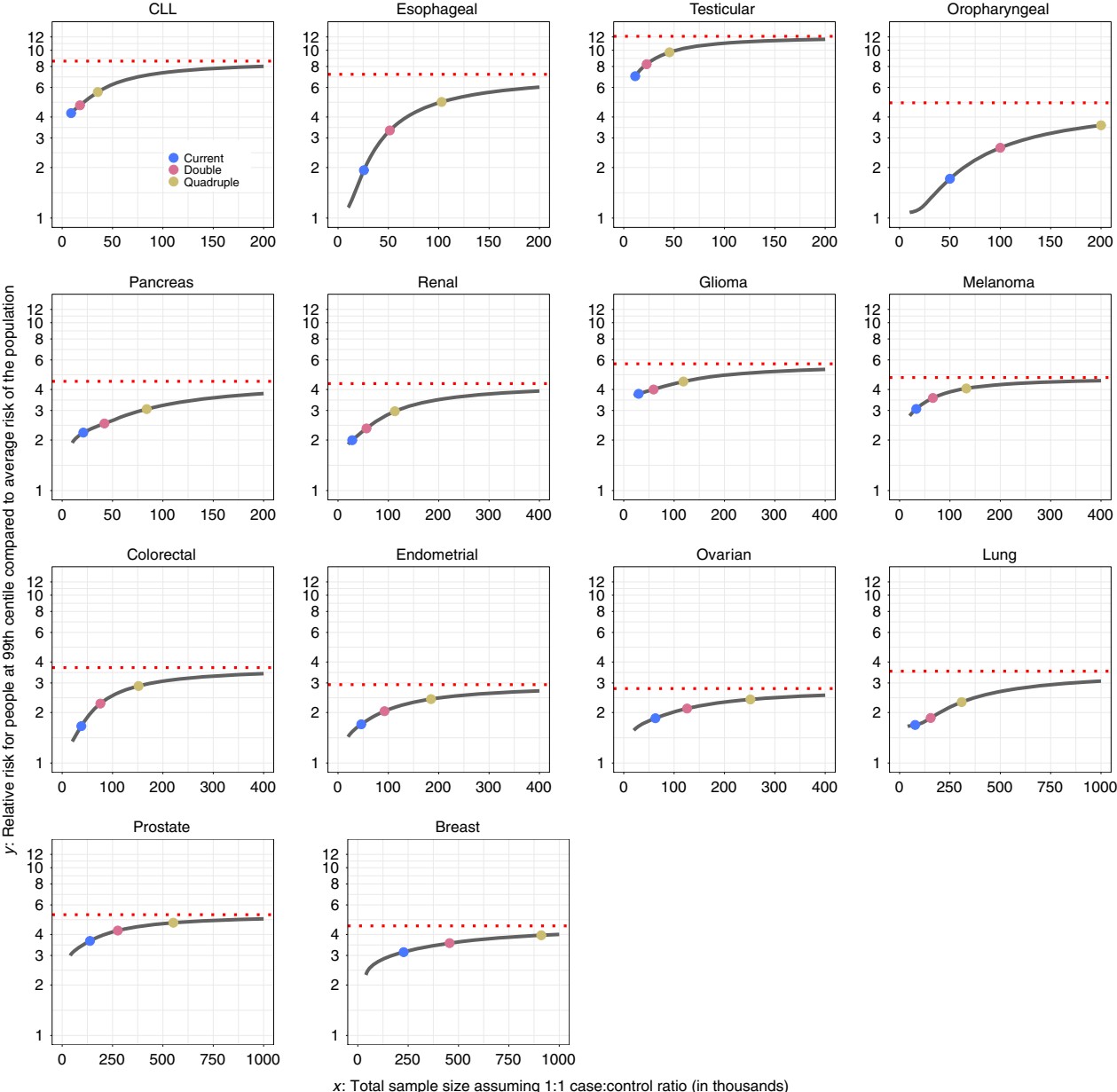

**Fig. 4 Projections of relative risks for individuals at or higher than 99th percentile of PRS as sample size for GWAS increases.** Results are shown where PRS is built based on SNPs at optimized *p* value threshold. The dotted horizontal red line indicates the maximum relative risk achievable according to estimate of GWAS heritability. Colored dots correspond to sample size for the largest published GWAS and those for doubled and quadruped sizes. *y*-Axis is presented in log10 scale. For oropharyngeal cancer, the projections at the "current sample size" are based on a sample size of 25K cases and 25K controls. For breast and esophageal cancer, the projections at the "current sample size" are based on the current largest GWAS sample sizes: 123K cases and 106K controls and 10K cases and 17K controls, respectively. For all other cancer sites, the projections at the "current sample size" are based on the GWAS sample sizes in Supplementary Table 1. CLL chronic lymphocytic leukemia.

etiologic subtypes. It is also noteworthy that testicular cancer also stands out for a large number of discoveries in cross-tissue expression quantitative trait loci analyses, likely indicating a stronger association of SNPs on gene expression levels for this tissue compared to others[41].

In conclusion, our comprehensive analysis of 14 cancer sites in adults of European ancestry reveals that, while all sites have polygenic influences, there is substantial diversity observed in their underlying genetic architectures, which reflects important biology and also influences the utility of polygenic risk prediction for individual cancers. Our projections for future yields of GWAS across these cancers provide a roadmap for important returns from future investment in research, including the potential

clinical utility of polygenic risk prediction for stratification of absolute risks in the population.

## Methods

**Description of GWAS studies**. We analyzed summary data from GWAS studies across 14 cancer types. For select cancer sites[26,28], we downloaded publicly available genome-wide summary-level statistics from the latest consortium-based analyses. For others, we obtained access to data through collaborative efforts with individual consortia. Details about individual studies, including the number of cases and controls, are provided in Supplementary Table 1.

**Linkage disequilibrium (LD) reference panel selection**. We consider a reference panel with ~1.07 million SNPs included in the HapMap3 and that had MAF > 0.05 in the 1000 Genome European Ancestry sample. Based on known LD among

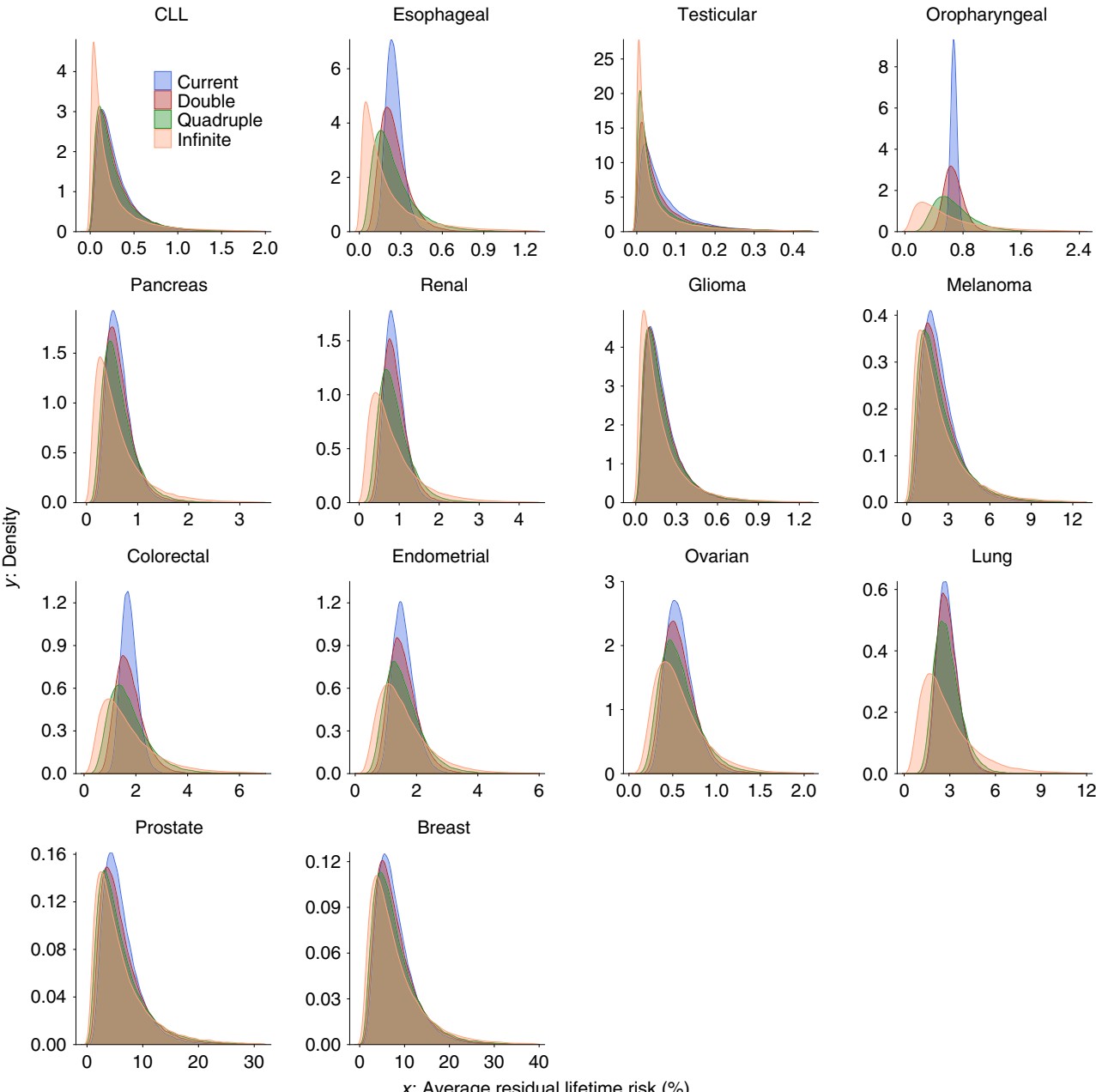

**Fig. 5 Projected distribution of average residual lifetime risk in the US population of non-Hispanic whites aged 30–75 years.** The risk is obtained according to variation of polygenic risk scores. The projections are shown for PRS built based on GWAS with current, doubled and quadrupled sample sizes and the best PRS that corresponds to limits defined by heritability. The projections are obtained by combining information on projected population variance of PRS, age-specific population incidence rate, competing risk of mortality and current distribution of age according to US 2016 census. For oropharyngeal cancer, the projections at the "current sample size" are based on a sample size of 25K cases and 25K controls. For breast and esophageal cancer, the projections at the "current sample size" are based on the current largest GWAS sample sizes: 123K cases and 106K controls and 10K cases and 17K controls, respectively. For all other cancer sites, the projections at the "current sample size" are based on the GWAS sample sizes in Supplementary Table 1. CLL chronic lymphocytic leukemia.

common variants, we expect these set of variants to provide high coverage for all common variants for European ancestry population and thus loss of information due to imperfect tagging of causal variants to be fairly minimal.

**Quality control for summary GWAS data**. Across all cancers, we applied several filtering steps analogous to those used earlier for estimation of heritability[45,46] and effect-size distribution using summary-level data[15]. First, we restricted analysis to SNPs within a set of reference ~1.07 million SNPs included in the HapMap3 and that had MAF > 0.05 in the 1000 Genome European Ancestry sample. Second, we excluded SNPs having substantial amounts of missing genotype data: sample sizes <0.67 times the 90th percentile of the distribution of sample sizes across all SNPs.

Third, we excluded SNPs within the major histocompatibility complex region (i.e., SNPs between 26,000,000 and 34,000,000 base pairs on chromosome six), which is known to have very complex allelic architecture and can have uncharacteristically large effects on some traits. Fourth, we removed regions that have SNPs with extremely large effect sizes to reduce possible undue influence of them on estimation of parameters associated with overall effect-size distributions. Using PLINK --clump, we identify all top SNPs that have associated chi-square statistics >80 (i.e., odds ratio (in standardized scale) >2.19) and removed all SNPs that were within 1-MB distance of or had an estimated squared LD >0.1 with those top SNPs. We added back the contribution of these top independent SNPs in the final reporting of the total number of susceptibility SNPs, estimates of total heritability, and various projections we made as a function of sample size of the GWAS.

**Statistical model**. We inferred common variant genetic architecture of the different cancers using GENESIS[15], a method we recently developed to characterize underlying effect-size distributions in terms of the total number of susceptibility SNPs (polygenicity) and a normal mixture model for the distribution of their effects. Specifically, it is assumed that standardized effects of common SNPs in an underlying logistic regression model on the risk of a cancer can be specified in the mixture distribution in the form $\beta_m \sim (1 - \pi_c)\delta_0 + \pi_c N(0, \sigma^2)$ (two-component model) or $\beta_m \sim (1 - \pi_c)\delta_0 + \pi_c [p_1 N(0, \sigma_1^2) + p_2 N(0, \sigma_2^2)]$ (three-component model) where $\delta_0$ is the Dirac delta function indicating that a fraction, $1 - \pi_c$, of the SNPs have null effects and remaining $\pi_c$ fraction of SNPs have non-null effects. Under the three-component model, $p_2 = 1 - p_1$ denotes the proportion of SNPs allocated to mixture component with larger variance component (assuming $\sigma_2^2 > \sigma_1^2$) models. Under these models, $M\pi_c$ characterizes the degree of polygenicity, i.e., the number of susceptibility SNPs with independent effects on disease risk. Under both models, we defined "GWAS heritability" of a disease as $h^2 = M\pi_c E(\beta^2)$, where $E(\beta^2)$ denotes the average variance size of the non-null SNPs. We observed that, under the above model, $h^2$ is also the population variance of the underlying "true" PRS, defined as $\text{PRS} = \sum_{m=1}^{M} \beta_m G_m$, where $G_m$ denotes the standardized genotype associated with the $m$th SNP. Under the two-component model, which assumes a single normal distribution for the effect of all susceptibility SNPs, $E(\beta^2) = \sigma^2$. Under the three-component model, which allows mixture of two normal distributions with distinct variance components and thus can better accommodate the presence of a group of susceptibility SNPs with much larger effects than others, we have $p_1\sigma_1^2 + p_2\sigma_2^2$. Under the three-component model, we use the fraction $v = p_2\sigma_2^2/(p_1\sigma_1^2 + p_2\sigma_2^2)$ to characterize the proportion of heritability explained by SNPs associated with the larger variance component parameter. As we removed SNPs with extremely large effects ($\chi_i^2 > 80$) and the associated regions from the analysis, in reporting the final heritability estimates, we added back the contribution of the independent top SNPs from these excluded regions as $\sum_i(\hat{\beta}_i^2 - \tau_i^2)$ where $\hat{\beta}_i$ is the estimate of log odds ratio (in standardized scale) and $\tau_i$ is the corresponding standard error for the $i$th SNP.

**Genetic variance projection**. Given the estimated effect-size distribution, we calculated expected discoveries and genetic variance explained using ED =

$M\hat{\pi}_c \int_\beta \text{pow}_{\alpha,n}(\beta) \sum_{h=1}^{H} \hat{p}_h N(0, \hat{\sigma}_h^2) d\beta$ and

$\text{EV} = M\hat{\pi}_c \int_\beta \beta^2 \text{pow}_{\alpha,n}(\beta) \sum_{h=1}^{H} \hat{p}_h N(0, \hat{\sigma}_h^2) d\beta$, respectively, at $\alpha = 5 \times 10^{-8}$ for a GWAS of sample size $n$, where $\text{pow}_{\alpha,n}(\beta) = 1 - \Phi\left(c_{\frac{\alpha}{2}} - \sqrt{n}\beta\right) + \Phi\left(-c_{\frac{\alpha}{2}} - \sqrt{n}\beta\right)$ with $\Phi(\cdot)$ the standard normal cumulative density function and $c_\alpha = \Phi^{-1}(1 - \alpha)$ the $\alpha$th quantile for the standard normal distribution. Similar to heritability calculations, we added back the contributions of independent top SNPs with very large effects to the number of expected discoveries and associated variances explained by the quantities $\sum_i \text{pow}_{\alpha,n}(\hat{\beta}_i)$ and $h^{-2}\sum_i(\hat{\beta}_i^2 - \tau_i^2)\text{pow}_{\alpha,n}(\hat{\beta}_i)$. We observed that for projections involving sample sizes bigger than the current study $\text{pow}_{\alpha,n}(\hat{\beta}_i)$ for the large effect SNPs will all be very close to 1.0.

**Projection for AUC and relative risk at top 1%**. As we quantify heritability in terms of the variability of the underlying "true" PRS, we used the formula[12,47,48] $\text{AUC} = \Phi\left(\sqrt{\frac{h^2}{2}}\right)$ to characterize the best discriminatory power achievable in limiting using common variant PRS. We used the same formula to calculate the AUC associated with PRSs that could be built using SNPs either reaching genome-wide significance ($p$ value $<5 \times 10^{-8}$) or a weaker but optimized threshold for a GWAS of given sample size based on the projected variance of the respective PRS. Given sample size of GWAS and an effect-size distribution for the underlying cancer, an optimal threshold for SNP selection that will maximize the expected predictive performance of PRS is calculated using analytic formula we have derived earlier[48]. The relative risk for those estimated to be at the 99th percentile or higher of the distribution of a PRS (compared to the average risk of the population) was calculated using the formula[12] $\exp\left(-\frac{h^2}{2} + \Phi^{-1}(0.99)\sqrt{h^2}\right)$, where $h^2$ is the population variance of the PRS.

**Absolute risk projection**. For each cancer site, we projected the distribution of residual lifetime risk (up to age 80 years) for non-Hispanic white individuals in the general US population according to PRSs, which could be built from GWASs of different sample sizes. For any given age, we first obtain the distribution of residual lifetime risk based on a model for absolute risks developed using the iCARE tool that we have described earlier[12,29]. The iCARE tool uses projected standard deviations of PRS at different GWAS sample sizes and age-specific cancer incidence rates available from the US National Cancer Institute-Surveillance, Epidemiology, and End Results Program (NCI-SEER) (2015) to obtain absolute risk distributions. In deriving absolute risks, we adjusted for competing risk of mortality due to other causes using the age-specific mortality rates from the Center for Disease Control WONDER database (2016). We then weighted the projected residual lifetime risk

distribution at different baseline ages (in 5-year categories) based on the US population distribution of ages within 30–75 years, as observed in the estimated 2016 US Census. For cancers of the reproductive system, weights were based on the age distributions among males or females, as appropriate.

**Reporting summary**. Further information on research design is available in the Nature Research Reporting Summary linked to this article.

## Data availability
The data that support the findings of this study are available by application from the participating consortia: BCAC (bcac@medschl.cam.ac.uk), BEACON (P Gharahkhani), ColonCFR (M Jenkins), GECCO/CORECT (U Peters), ECAC (TA O'Mara), GenoMEL (M Iles), GICC (R Houlston), ILLCO/INTEGRAL (C Amos), InterLymph (S Berndt), OCAC (PDP Pharoah), Oral Cancer GWAS (P Brennan), PanC4/PanScan (LT Amundadottir), PRACTICAL (Data Access Committee/http://practical.icr.ac.uk/), Renal Cancer GWAS (MP Purdue, P Brennan), and TECAC (KA McGlynn). For breast and prostate cancers, summary GWAS data can also be downloaded from http://bcac.ccge. medschl.cam.ac.uk/bcacdata/oncoarray/gwas-icogs-and-oncoarray-summary-results/ and http://practical.icr.ac.uk/blog/?page_id=8164.

## Code availability
The code for running the analysis in the paper is freely available from the CancerEffectSize GitHub repository (https://github.com/yandorazhang/CancerEffectSize).

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

## Acknowledgements

The research was supported by an RO1 grant from NHGRI (1 R01 HG010480-01) and the intramural program of the National Cancer Institute.

## Author contributions

N.C. and M.G.-C. conceived the project. Y.Z. and A.W. performed main analyses. Y.Z., N.C., and M.G.-C. wrote the first draft of the manuscript. BCAC, BEACON, CCFR, CORECT, ECAC, GECCO, GenoMEL, GICC, ILCCO, Integral, InterLymph, OCAC, Oral Cancer GWAS, PANC4, PanScan, PRACTICAL, Renal Cancer GWAS, and TECAC contributed data. P.P.C., R.L.M., M.K.S., M.J., U.P., L.H., S.L. Schmit., T.A.O., A.B.S., D.J. T., M.H.L., M.M.I., F.D., S.M., S.V.W., M.R.W., C.I.A., S.I.B., B.M.B., N.J.C., P.D.P.P., T. A.S., L.T.A., E.J.J., H.A.R., R.Z.S.-S., M.P.P., M.H.G., K.A.M., and S.J.C. commented on earlier drafts of the manuscript. H.Z., D.F.E., J.S., P.H., K.M., J.D., J.C.-C., P.G., D.W., P. T.C., M.H., S.B.G., G.C., I.T., I.D.V., M.T.L., R.K., D.T.B., M.L.B., R.H., J.K.W., B.M., J.B.-S., B.K., R.J.H., P.B., J.M., N.E.C., P.K., N.R., S.L. Slager., A.B., S.A.G., C.L.P., E.L.G., J.M. S., K.B.M., A.P.K., G.M.P., B.M.W., D.L., R.A.E., C.A.H., Z.K.-J., F.R.S., A.A.A.O., G.S., M.D.D., T.G., P.A.K., K.L.N., C.T., and F.W. reviewed the manuscript. All authors reviewed and approved the final draft of the manuscript.

## Competing interests

The authors declare no competing interests.

## Additional information

Yan Dora Zhang 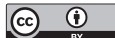[1,2], Amber N. Hurson [3,4], Haoyu Zhang[3,5], Parichoy Pal Choudhury[3], Douglas F. Easton [6,7], Roger L. Milne [8,9,10], Jacques Simard [11], Per Hall[12,13], Kyriaki Michailidou [7,14], Joe Dennis [7],

Marjanka K. Schmidt[15,16], Jenny Chang-Claude[17,18], Puya Gharahkhani[19], David Whiteman[20], Peter T. Campbell[21], Michael Hoffmeister[22], Mark Jenkins[9], Ulrike Peters[23], Li Hsu[23], Stephen B. Gruber[24], Graham Casey[25], Stephanie L. Schmit[26], Tracy A. O'Mara[27], Amanda B. Spurdle[27], Deborah J. Thompson[7], Ian Tomlinson[28,29], Immaculata De Vivo[30,31], Maria Teresa Landi[3], Matthew H. Law[19], Mark M. Iles[32], Florence Demenais[33], Rajiv Kumar[34], Stuart MacGregor[19], D. Timothy Bishop[35], Sarah V. Ward[36], Melissa L. Bondy[37], Richard Houlston[38], John K. Wiencke[39], Beatrice Melin[40], Jill Barnholtz-Sloan[41], Ben Kinnersley[38], Margaret R. Wrensch[39], Christopher I. Amos[42], Rayjean J. Hung[43], Paul Brennan[44], James McKay[44], Neil E. Caporaso[3], Sonja I. Berndt[3], Brenda M. Birmann[30], Nicola J. Camp[45], Peter Kraft[46], Nathaniel Rothman[3], Susan L. Slager[47], Andrew Berchuck[48], Paul D. P. Pharoah[6,7], Thomas A. Sellers[26], Simon A. Gayther[49], Celeste L. Pearce[24,50], Ellen L. Goode[51], Joellen M. Schildkraut[52], Kirsten B. Moysich[53], Laufey T. Amundadottir[54], Eric J. Jacobs[21], Alison P. Klein[55], Gloria M. Petersen[51], Harvey A. Risch[56], Rachel Z. Stolzenberg-Solomon[3], Brian M. Wolpin[57], Donghui Li[58], Rosalind A. Eeles[59], Christopher A. Haiman[24], Zsofia Kote-Jarai[59], Fredrick R. Schumacher[60], Ali Amin Al Olama[61,62], Mark P. Purdue[3], Ghislaine Scelo[44], Marlene D. Dalgaard[63,64], Mark H. Greene[65], Tom Grotmol[66], Peter A. Kanetsky[26], Katherine A. McGlynn[3], Katherine L. Nathanson[67], Clare Turnbull[38], Fredrik Wiklund[68], Breast Cancer Association Consortium (BCAC)*, Barrett's and Esophageal Adenocarcinoma Consortium (BEACON)*, Colon Cancer Family Registry (CCFR)*, Transdisciplinary Studies of Genetic Variation in Colorectal Cancer (CORECT)*, Endometrial Cancer Association Consortium (ECAC)*, Genetics and Epidemiology of Colorectal Cancer Consortium (GECCO)*, Melanoma Genetics Consortium (GenoMEL)*, Glioma International Case-Control Study (GICC)*, International Lung Cancer Consortium (ILCCO) *, Integrative Analysis of Lung Cancer Etiology and Risk (INTEGRAL) Consortium*, International Consortium of Investigators Working on Non-Hodgkin's Lymphoma Epidemiologic Studies (InterLymph)*, Ovarian Cancer Association Consortium (OCAC)*, Oral Cancer GWAS*, Pancreatic Cancer Case-Control Consortium (PanC4)*, Pancreatic Cancer Cohort Consortium (PanScan)*, Prostate Cancer Association Group to Investigate Cancer Associated Alterations in the Genome (PRACTICAL)*, Renal Cancer GWAS*, Testicular Cancer Consortium (TECAC)*, Stephen J. Chanock[3], Nilanjan Chatterjee[5,55,69]✉ & Montserrat Garcia-Closas[3,69]

[1]Department of Statistics and Actuarial Science, Faculty of Science, The University of Hong Kong, Hong Kong SAR, China. [2]Centre for PanorOmic Sciences, Li Ka Shing Faculty of Medicine, The University of Hong Kong, Hong Kong SAR, China. [3]Division of Cancer Epidemiology and Genetics, National Cancer Institute, Rockville, MD, USA. [4]Department of Epidemiology, Gillings School of Global Public Health, University of North Carolina at Chapel Hill, Chapel Hill, NC, USA. [5]Department of Biostatistics, Bloomberg School of Public Health, Johns Hopkins University, Baltimore, MD, USA. [6]Department of Oncology, Centre for Cancer Genetic Epidemiology, University of Cambridge, Cambridge, UK. [7]Department of Public Health and Primary Care, Centre for Cancer Genetic Epidemiology, University of Cambridge, Cambridge, UK. [8]Cancer Epidemiology Division, Cancer Council Victoria, Melbourne, VIC, Australia. [9]Centre for Epidemiology and Biostatistics, Melbourne School of Population and Global Health, The University of Melbourne, Melbourne, VIC, Australia. [10]Precision Medicine, School of Clinical Sciences at Monash Health, Monash University, Clayton, VIC, Australia. [11]Centre Hospitalier Universitaire de Québec–Université Laval Research Center, Québec City, QC, Canada. [12]Department of Medical Epidemiology and Biostatistics, Karolinska Institutet, Stockholm, Sweden. [13]Department of Oncology, Södersjukhuset, Stockholm, Sweden. [14]Department of Electron Microscopy/Molecular Pathology and The Cyprus School of Molecular Medicine, The Cyprus Institute of Neurology & Genetics, Nicosia, Cyprus. [15]Division of Molecular Pathology, The Netherlands Cancer Institute - Antoni van Leeuwenhoek Hospital, Amsterdam, The Netherlands. [16]Division of Psychosocial Research and Epidemiology, The Netherlands Cancer Institute - Antoni van Leeuwenhoek Hospital, Amsterdam, The Netherlands. [17]Division of Cancer Epidemiology, German Cancer Research Center (DKFZ), Heidelberg, Germany. [18]Cancer Epidemiology Group, University Cancer Center Hamburg (UCCH), University Medical Center Hamburg-Eppendorf, Hamburg, Germany. [19]Statistical Genetics, QIMR Berghofer Medical Research Institute, Brisbane, QLD, Australia. [20]Cancer Control, QIMR Berghofer Medical Research Institute, Brisbane, QLD, Australia. [21]Behavioral and Epidemiology Research Group, American Cancer Society, Atlanta, GA, USA. [22]Division of Clinical Epidemiology and Aging Research, German Cancer Research Center (DKFZ), Heidelberg, Germany. [23]Public Health Sciences Division, Fred Hutchinson Cancer Research Center, Seattle, WA, USA. [24]Department of Preventive Medicine, USC Norris Comprehensive Cancer Center, Keck School of Medicine, University of Southern California, Los Angeles, CA, USA. [25]Department of Public Health Sciences, Center for Public Health Genomics, University of Virginia, Charlottesville, VA, USA. [26]Department of Cancer Epidemiology, H. Lee Moffitt Cancer Center and Research Institution, Tampa, FL, USA. [27]Genetics and Computational Biology Division, QIMR Berghofer Medical Research Institute, Brisbane, QLD, Australia. [28]Institute of Cancer and Genomic Sciences, University of Birmingham, Birmingham, UK. [29]Wellcome Trust Centre for Human Genetics and Oxford NIHR Biomedical Research Centre, University of Oxford, Oxford, UK. [30]Channing Division of Network Medicine, Department of Medicine, Brigham and Women's Hospital and Harvard Medical School, Boston, MA, USA. [31]Department of Epidemiology, Harvard T.H. Chan School of

Public Health, Boston, MA, USA. [32]Section of Epidemiology and Biostatistics, Leeds Institute of Cancer and Pathology, University of Leeds, Leeds, UK. [33]Université de Paris, UMRS-1124, Institut National de la Santé et de la Recherche Médicale (INSERM), 75006 Paris, France. [34]Division of Molecular Genetic Epidemiology, German Cancer Research Center (DKFZ), Heidelberg, Germany. [35]Division of Haematology and Immunology, Leeds Institute of Medical Research, University of Leeds, Leeds, UK. [36]Centre for Genetic Origins of Health and Disease, School of Biomedical Sciences, The University of Western Australia, Perth, WA, Australia. [37]Department of Medicine, Section of Epidemiology and Population Sciences, Baylor College of Medicine, Houston, TX, USA. [38]Division of Genetics and Epidemiology, The Institute of Cancer Research, London, UK. [39]Department of Neurological Surgery, School of Medicine, University of California, San Francisco, San Francisco, CA, USA. [40]Department of Radiation Sciences Oncology, Umeå University, Umeå, Sweden. [41]Case Comprehensive Cancer Center, Case Western Reserve University School of Medicine, Cleveland, OH, USA. [42]Institute for Clinical and Translational Research, Dan L. Duncan Comprehensive Cancer Center, Baylor College of Medicine, Houston, TX, USA. [43]Lunenfeld-Tanenbuaum Research Institute, Sinai Health System, Toronto, ON, Canada. [44]International Agency for Research on Cancer, World Health Organization, Lyon, France. [45]Division of Hematology and Hematological Malignancies, University of Utah School of Medicine, Salt Lake City, UT, USA. [46]Program in Genetic Epidemiology and Statistical Genetics, Harvard T.H. Chan School of Public Health, Boston, MA, USA. [47]Division of Biomedical Statistics & Informatics, Department of Health Sciences Research, Mayo Clinic, Rochester, MN, USA. [48]Department of Gynecologic Oncology, Duke University Medical Center, Durham, NC, USA. [49]Center for Bioinformatics and Functional Genomics and the Cedars Sinai Genomics Core, Cedars-Sinai Medical Center, Los Angeles, CA, USA. [50]Department of Epidemiology, University of Michigan School of Public Health, Ann Arbor, MI, USA. [51]Division of Epidemiology, Department of Health Science Research, Mayo Clinic, Rochester, MN, USA. [52]Rollins School of Public Health, Emory University, Atlanta, GA, USA. [53]Division of Cancer Prevention and Control, Roswell Park Cancer Institute, Buffalo, NY, USA. [54]Laboratory of Translational Genomics, Division of Cancer Epidemiology and Genetics, National Cancer Institute, National Institutes of Health, Bethesda, MD, USA. [55]Department of Oncology, Sidney Kimmel Comprehensive Cancer Center, Johns Hopkins School of Medicine, Baltimore, MD, USA. [56]Chronic Disease Epidemiology, Yale School of Medicine, New Haven, CT, USA. [57]Department of Medical Oncology, Dana-Farber Cancer Institute, Boston, MA, USA. [58]Division of Cancer Medicine, GI Medical Oncology Department, The University of Texas MD Anderson Cancer Center, Houston, TX, USA. [59]Division of Genetics and Epidemiology, The Institute of Cancer Research, Sutton, Surrey, UK. [60]Department of Population and Quantitative Health Sciences, Case Western Reserve University School of Medicine, Cleveland, OH, USA. [61]Strangeways Research Laboratory, Department of Public Health and Primary Care, Centre for Cancer Genetic Epidemiology, University of Cambridge, Cambridge, UK. [62]Department of Clinical Neurosciences, University of Cambridge, Cambridge, UK. [63]Department of Growth and Reproduction, Copenhagen University Hospital (Rigshospitalet), Copenhagen, Denmark. [64]Department of Health Technology, Technical University of Denmark, Lyngby, Denmark. [65]Clinical Genetics Branch, Division of Cancer Genetics and Epidemiology, National Cancer Institute, Rockville, MD, USA. [66]Cancer Registry of Norway, Oslo, Norway. [67]Division of Translational Health and Human Genetics, Department of Medicine, University of Pennsylvania, Philadelphia, PA, USA. [68]Department of Medical Epidemiology and Biostatistics, Karolinska Institutet, Stockholm, Sweden. [69]These authors contributed equally: Nilanjan Chatterjee, Montserrat Garcia-Closas *Lists of authors and their affiliations appear at the end of the paper. ✉email: nchatte2@jhu.edu

## Breast Cancer Association Consortium (BCAC)

Douglas F. Easton[6,7], Roger L. Milne[8,9,10], Jacques Simard[11], Per Hall[12,13], Kyriaki Michailidou[7,14], Joe Dennis[7], Marjanka K. Schmidt[15,16] & Jenny Chang-Claude[17,18]

## Barrett's and Esophageal Adenocarcinoma Consortium (BEACON)

Puya Gharahkhani[19] & David Whiteman[20]

## Colon Cancer Family Registry (CCFR)

Peter T. Campbell[21], Michael Hoffmeister[22], Mark Jenkins[9], Ulrike Peters[23], Li Hsu[23], Stephen B. Gruber[24], Graham Casey[25] & Stephanie L. Schmit[26]

## Transdisciplinary Studies of Genetic Variation in Colorectal Cancer (CORECT)

Peter T. Campbell[21], Michael Hoffmeister[22], Mark Jenkins[9], Ulrike Peters[23], Li Hsu[23], Stephen B. Gruber[24], Graham Casey[25] & Stephanie L. Schmit[26]

## Endometrial Cancer Association Consortium (ECAC)

Tracy A. O'Mara[27], Amanda B. Spurdle[27], Deborah J. Thompson[7], Ian Tomlinson[28,29] & Immaculata De Vivo[30,31]

## Genetics and Epidemiology of Colorectal Cancer Consortium (GECCO)

Peter T. Campbell[21], Michael Hoffmeister[22], Mark Jenkins[9], Ulrike Peters[23], Li Hsu[23], Stephen B. Gruber[24], Graham Casey[25] & Stephanie L. Schmit[26]

**Melanoma Genetics Consortium (GenoMEL)**

Maria Teresa Landi[3], Matthew H. Law[19], Mark M. Iles[32], Florence Demenais[33], Rajiv Kumar[34], Stuart MacGregor[19], David T. Bishop[35] & Sarah V. Ward[36]

**Glioma International Case-Control Study (GICC)**

Melissa L. Bondy[37], Richard Houlston[38], John K. Wiencke[39], Beatrice Melin[40], Jill Barnholtz-Sloan[41], Ben Kinnersley[38] & Margaret R. Wrensch[39]

**International Lung Cancer Consortium (ILCCO)**

Christopher I. Amos[42], Rayjean J. Hung[43], Paul Brennan[44], James McKay[44] & Neil E. Caporaso[3]

**Integrative Analysis of Lung Cancer Etiology and Risk (INTEGRAL) Consortium**

Christopher I. Amos[42], Rayjean J. Hung[43], Paul Brennan[44], James McKay[44] & Neil E. Caporaso[3]

**International Consortium of Investigators Working on Non-Hodgkin's Lymphoma Epidemiologic Studies (InterLymph)**

Sonja I. Berndt[3], Brenda M. Birmann[30], Nicola J. Camp[45], Peter Kraft[46], Nathaniel Rothman[3] & Susan L. Slager[47]

**Ovarian Cancer Association Consortium (OCAC)**

Andrew Berchuck[48], Paul D. P. Pharoah[6,7], Thomas A. Sellers[26], Simon A. Gayther[49], Celeste L. Pearce[24,50], Ellen L. Goode[51], Joellen M. Schildkraut[52] & Kirsten B. Moysich[53]

**Oral Cancer GWAS**

Christopher I. Amos[42], Paul Brennan[44] & James McKay[44]

**Pancreatic Cancer Case-Control Consortium (PanC4)**

Laufey T. Amundadottir[54], Eric J. Jacobs[21], Alison P. Klein[55], Gloria M. Petersen[51], Harvey A. Risch[56], Rachel Z. Stolzenberg-Solomon[3], Brian M. Wolpin[57] & Donghui Li[58]

**Pancreatic Cancer Cohort Consortium (PanScan)**

Laufey T. Amundadottir[54], Eric J. Jacobs[21], Alison P. Klein[55], Gloria M. Petersen[51], Harvey A. Risch[56], Rachel Z. Stolzenberg-Solomon[3], Brian M. Wolpin[57] & Donghui Li[58]

**Prostate Cancer Association Group to Investigate Cancer Associated Alterations in the Genome (PRACTICAL)**

Rosalind A. Eeles[59], Christopher A. Haiman[24], Zsofia Kote-Jarai[59], Fredrick R. Schumacher[60] & Ali Amin Al Olama[61,62]

**Renal Cancer GWAS**

Mark P. Purdue[3] & Ghislaine Scelo[44]

## Testicular Cancer Consortium (TECAC)

Marlene D. Dalgaard[63,64], Mark H. Greene[65], Tom Grotmol[66], Peter A. Kanetsky[26], Katherine A. McGlynn[3], Katherine L. Nathanson[67], Clare Turnbull[38] & Fredrik Wiklund[68]

A full list of members and their affiliations appears in the Supplementary Information.

