## [Peer Review File · Nature Communications]

Reviewers' comments:

Reviewer #1 (Remarks to the Author):

This is a well-written article describing an analysis of 14 GWAS studies of various cancers with the goal of estimating the degree of polygenicity and the effect size distribution of cancer associated SNPs. This is important because we have already observed that different cancers have more or less GWAS SNPs and some SNPs are associated with higher risks. The investigators have employed their recently published method to accomplish the analysis using summary statistics from previously conducted studies. The results are intriguing and suggest that PRS may be useful for two sets of cancers: those with SNPs demonstrating large effect sizes (eg testicular cancer) and those with there are a large number of associated SNPs that explain heritability (eg breast cancer) Despite the hypothetical nature of this work, the results are based on large datasets and there are clinical implications for risk modeling that are intuitive.

There is no stated hypothesis for this work, this might be helpful for the presentation.

Reviewer #2 (Remarks to the Author):

Title: "Assessment of Polygenic Architecture and Risk Prediction based on Common Variants Across Fourteen Cancers"

Summary:

In their study Yan Zhang et al. explored summary statistics of 14 large, European ancestry-based cancer GWAS to estimate for each the number of contributing common risk variant (minor allele frequency above 5%) and their effect size distributions.

Through simulations they predicted the required GWAS sample sizes for each of the 14 cancers to identify the risk variants that together could explain 80% of the estimated chip heritability. Finally, they also projected for each of the 14 cancers the maximally achievable relative risk for individuals in the highest percentile of polygenic risk scores compared to the average risk.

It was a pleasure to read this well-crafted manuscript which represents a convincing contribution to the field of cancer GWAS. The authors highlight limitations and promising avenues for current and future GWAS and polygenic risk score applications in cancer.

Minor concerns:

* The summary-level data from genome-wide association studies (GWAS) were reduced to about 1.1 million SNPs that overlapped with HapMap3 SNPs and were common (minor allele frequency [MAF] > 5%). By doing so, the original number of available summary-level data was markedly reduced. For example, the two publicly available GWAS on breast and prostate cancer each include over 7 million common SNPs (MAF > 5%) and over 4 million low-frequency SNPs (MAF ≤ 5% and MAF > 0.5%) (Schumacher et al. 2018, Nature genetics, PMID: 29892016) and Michailidou et al. 2017, Nature, PMID 29059683). Also, chromosome X variants were likely excluded in this filtering step (1000 Genomes Project Consortium, Nature. 2010, PMID: 20981092).

A broad readership would benefit from a more transparent description of the SNP filtering process, e.g. by clearly stating the consequences of only including common HapMap3 SNPs (e.g. in the method section, Supplementary Table 1 [BEFORE filtering in the model], and the discussion).

Furthermore, these SNP exclusions might have influenced several of the presented estimates, e.g., the number of independent risk associations and the sample size projections of future GWAS. The authors should consider emphasising this in their discussion.

* Table 1 lists total heritability estimates of the top three cancers (CLL, oesophageal, and

testicular) that exceed 1. It doesn't seem intuitive to observe such high values. The authors might consider providing additional information on the scale and interpretation of these estimates.

* Methods: It's not clear how LD-clumping and P-value optimisation was performed. A description of the applied LD reference, software and methods would increase transparency.

* Interpretation of Figure 4 would benefit from a consistent y-axis, especially of the top row, because the current inconsistencies make direct comparisons difficult (y-axis of testicular cancer does not include 1).

* Supplementary Fig. 1 & 2: Moving the row labels (2- and 3-component) from the right to left side would improve readability.

Reviewers' comments:

Reviewer #1 (Remarks to the Author):

This is a well-written article describing an analysis of 14 GWAS studies of various cancers with the goal of estimating the degree of polygenicity and the effect size distribution of cancer associated SNPs. This is important because we have already observed that different cancers have more or less GWAS SNPs and some SNPs are associated with higher risks. The investigators have employed their recently published method to accomplish the analysis using summary statistics from previously conducted studies. The results are intriguing and suggest that PRS may be useful for two sets of cancers: those with SNPs demonstrating large effect sizes (eg testicular cancer) and those with there are a large number of associated SNPs that explain heritability (eg breast cancer)

Despite the hypothetical nature of this work, the results are based on large datasets and there are clinical implications for risk modeling that are intuitive.

There is no stated hypothesis for this work, this might be helpful for the presentation.

We would like to thank the reviewer for the encouraging comments. We also believe our contributions would be meaningful to the scientific community.

Reviewer #2 (Remarks to the Author):

Title: "Assessment of Polygenic Architecture and Risk Prediction based on Common Variants Across Fourteen Cancers"

Summary:

In their study Yan Zhang et al. explored summary statistics of 14 large, European ancestry-based cancer GWAS to estimate for each the number of contributing common risk variant (minor allele frequency above 5%) and their effect size distributions.

Through simulations they predicted the required GWAS sample sizes for each of the 14 cancers to identify the risk variants that together could explain 80% of the estimated chip heritability. Finally, they also projected for each of the 14 cancers the maximally achievable relative risk for individuals in the highest percentile of polygenic risk scores compared to the average risk.

It was a pleasure to read this well-crafted manuscript which represents a convincing contribution to the field of cancer GWAS. The authors highlight limitations and promising avenues for current and future GWAS and polygenic risk score applications in cancer.

We would like to thank the reviewer for the additional comments. We have now modified our manuscripts to address these comments. We feel the comments of the reviewer has improved the quality of the paper substantially.

Minor concerns:

* The summary-level data from genome-wide association studies (GWAS) were reduced to about 1.1

million SNPs that overlapped with HapMap3 SNPs and were common (minor allele frequency [MAF] > 5%). By doing so, the original number of available summary-level data was markedly reduced. For example, the two publicly available GWAS on breast and prostate cancer each include over 7 million common SNPs (MAF > 5%) and over 4 million low-frequency SNPs (MAF ≤ 5% and MAF > 0.5%) (Schumacher et al. 2018, Nature genetics, PMID: 29892016) and Michailidou et al. 2017, Nature, PMID 29059683). Also, chromosome X variants were likely excluded in this filtering step (1000 Genomes Project Consortium, Nature. 2010, PMID: 20981092).

A broad readership would benefit from a more transparent description of the SNP filtering process, e.g. by clearly stating the consequences of only including common HapMap3 SNPs (e.g. in the method section, Supplementary Table 1 [BEFORE filtering in the model], and the discussion).

Furthermore, these SNP exclusions might have influenced several of the presented estimates, e.g., the number of independent risk associations and the sample size projections of future GWAS. The authors should consider emphasising this in their discussion.

We thank the reviewer for the valuable comments. We have now added a paragraph describing the LD reference panel selection in the Methods section (page 12) to make it more clear about the decision to use only 1.1 million Hapmap3 common SNPs. Also, we have made the code of SNP filtering process as well as all other analysis publicly available in <https://github.com/yandorazhang/CancerEffectSize> (a code availability statement is added to the main paper).

* Table 1 lists total heritability estimates of the top three cancers (CLL, oesophageal, and testicular) that exceed 1. It doesn't seem intuitive to observe such high values. The authors might consider providing additional information on the scale and interpretation of these estimates.

The heritability reported in Table 1 is on the log-odds-ratio scale which can take value higher than 1.0 as this is not represented as a proportion of variance explained. In particular, as explained in the footnote of Table 1, the heritability is characterized by population variance of the underlying true PRS as $h^2 = \text{Var}(\sum_{m=1}^M \beta_m G_m) = M\pi_c E(\beta^2)$, where $E(\beta^2)$ denotes per-SNP effect-size of the non-null SNPs in the log-odds-ratio scale. We present heritability in this scale because this is the relevant quantity that directly relates to performance of PRS for risk-prediction. In the column heading of Table 1 we have now clarified that this is heritability in log-odds-ratio scale and also clarified it further in the footnote.

One can transform this heritability into liability scale, which takes values between 0 and 1 and are often reported in literatures, we can apply the equation $h_l^2 = h^2 K^2 (1 - K)^2 / z^2$, where K is the population prevalence for the disease, and $z = \phi(\Phi^{-1}(1 - K))$ where $\phi(\cdot)$ and $\Phi(\cdot)$ are the probability density and cumulative density function for standard normal distribution respectively.

* Methods: It's not clear how LD-clumping and P-value optimisation was performed. A description of the applied LD reference, software and methods would increase transparency.

We have now deposited the annotated R code for each of the steps of our analysis deposited in GitHub (source code link: <https://github.com/yandorazhang/CancerEffectSize>). In particular, we note that we use theoretical results we had derived earlier (Chatterjee et al., Nature Genetics, 2013) to derive

optimal p-value threshold for SNP selection at a given sample size and the estimated effect-size distribution for a trait. In the methods section (page 15) we have stated

“Given sample size of GWAS and an effect-size distribution for the underlying cancer, an optimal threshold for SNP selection that will maximize the expected predictive performance of PRS is calculated using analytic formula we have derive earlier⁴⁸”

Also we include the links for the data sources which are publicly available. For others, however, we do not have permission to share the summary-statistics data as part of this manuscript.

* Interpretation of Figure 4 would benefit from a consistent y-axis, especially of the top row, because the current inconsistencies make direct comparisons difficult (y-axis of testicular cancer does not include 1).

We thank the reviewer for the comments. We have now plotted the y-axis to logarithm10 scale for Figure 4 (and Supplementary Figure 4) so that all results can be shown in the same y-axis scale and yet the some figures will not be too squeezed.

* Supplementary Fig. 1 & 2: Moving the row labels (2- and 3-component) from the right to left side would improve readability.

We have modified the figures accordingly.

REVIEWERS' COMMENTS:

Reviewer #2 (Remarks to the Author):

I would like to thank Dr Chatterjee and his colleagues for revising their manuscript and satisfactorily addressing all the minor points that I raised in the previous round of review. In addition, the authors now freely shared their analysis code through a GitHub repository and by doing so further increased transparency and ensured that interested readers can reproduce their presented work.

I have no further questions and concerns about the paper.

REVIEWERS' COMMENTS:

Reviewer #2 (Remarks to the Author):

I would like to thank Dr Chatterjee and his colleagues for revising their manuscript and satisfactorily addressing all the minor points that I raised in the previous round of review. In addition, the authors now freely shared their analysis code through a GitHub repository and by doing so further increased transparency and ensured that interested readers can reproduce their presented work. I have no further questions and concerns about the paper.

We thank the reviewer for the comments to help improve our manuscript.